

# The behaviour and activity budgets of two sympatric sloths; *Bradypus variegatus* and *Choloepus hoffmanni*

Rebecca N. Cliffe[1,2,3], Ryan J. Haupt[4], Sarah Kennedy[1], Cerys Felton[2], Hannah J. Williams[2,5], Judy Avey-Arroyo[3] and Rory Wilson[2]

[1] The Sloth Conservation Foundation, Hayfield, Derbyshire, United Kingdom
[2] Swansea Lab for Animal Movement, Biosciences, College of Science, Swansea University, Swansea, United Kingdom
[3] The Sloth Sanctuary of Costa Rica, Limon, Costa Rica
[4] Department of Geology and Geophysics, University of Wyoming, Laramie, WY, United States of America
[5] Migration Department, Max Planck Institute of Animal Behavior, Radolfzell, Germany

Corresponding author
Rebecca N. Cliffe,
rebecca@slothconservation.org

## ABSTRACT

It is usually beneficial for species to restrict activity to a particular phase of the 24-hour cycle as this enables the development of morphological and behavioural adaptations to enhance survival under specific biotic and abiotic conditions. Sloth activity patterns are thought to be strongly related to the environmental conditions due to the metabolic consequences of having a low and highly variable core body temperature. Understanding the drivers of sloth activity and their ability to withstand environmental fluctuations is of growing importance for the development of effective conservation measures, particularly when we consider the vulnerability of tropical ecosystems to climate change and the escalating impacts of anthropogenic activities in South and Central America. Unfortunately, the cryptic nature of sloths makes long term observational research difficult and so there is very little existing literature examining the behavioural ecology of wild sloths. Here, we used micro data loggers to continuously record, for the first time, the behaviour of both *Bradypus* and *Choloepus* sloths over periods of days to weeks. We investigate how fluctuations in the environmental conditions affect the activity of sloths inhabiting a lowland rainforest on the Caribbean coast of Costa Rica and examined how this might relate to their low power lifestyle. Both *Bradypus* and *Choloepus* sloths were found to be cathemeral in their activity, with high levels of between-individual and within-individual variation in the amounts of time spent active, and in the temporal distribution of activity over the 24-hour cycle. Daily temperature did not affect activity, although *Bradypus* sloths were found to show increased nocturnal activity on colder nights, and on nights following colder days. Our results demonstrate a distinct lack of synchronicity within the same population, and we suggest that this pattern provides sloths with the flexibility to exploit favourable environmental conditions whilst reducing the threat of predation.

## INTRODUCTION

For the majority of animals, time is clearly divided between periods of activity and periods of inactivity (*Halle & Stenseth, 2000*). During active periods, animals are considered to allocate time to different behaviours judiciously to maximize lifetime reproductive success (*Pianka, 1976*). All behaviours require energy though, with the combined sum of the power costs of an animal's daily energy budget modulating the animal's field metabolic rate (FMR) (*Russell et al., 2015*). Successful animals behave so as to minimize energy expenditure and maximize energetic return (maximizing 'net energy gain') (*Pyke, Pulliam & Charnov, 1977*; *Stephens & Krebs, 1986*), ultimately investing the gain in resources into reproduction (*Alcock, 2009*).

Activity patterns vary greatly across and within taxonomic groups, with different taxa being primarily diurnal, nocturnal, crepuscular or, occasionally, a combination of all these (*Ashby, 1972*). Species that are able to operate across all diel phases include those animals with multi-modal rhythms of activity (including poly-phasic and ultradian rhythms) (*Halle, 2006*), and cathemeral species which show significant activity during both the light and dark phases of the 24-hour cycle (*Tattersall, 1987*). In such species, the temporal distribution of activity is often flexible, being governed by complex interactions that occur between the animals' endogenous rhythms, ecological entrainment mechanisms, and various environmental or ecological factors such as ambient temperature, light, food availability, interspecific competition, and predation risk (*Donati & Borgognini-Tarli, 2006*; *Grignolio et al., 2018*; *Perea-Rodríguez et al., 2022*). It is considered that cathemerality in particular arises as a consequence of an animal's true endogenous rhythm being 'masked' by these external factors (*Curtis & Rasmussen, 2006*). Cathemerality can encompass regular activity spanning the 24-hour cycle year-round, or can occur on a more cyclic basis, shifting from nocturnality to diurnality between different days or seasons (*Hofmann et al., 2016*; *Perea-Rodríguez et al., 2022*; *Van der Vinne et al., 2014*). However, details of the complex biotic and abiotic factors that govern such flexible activity patterns remain poorly understood (*Halle, 2000*; *Kappeler & Erkert, 2003*).

Sloths (Genera: *Bradypus* and *Choloepus*) are cryptic canopy mammals that are considered to be cathemeral and nocturnal in their activity patterns, respectively (*Chiarello, 1998a*; *Gilmore, Da Costa & Duarte, 2001*; *Giné et al., 2015*). However, due to their slow movements (*Sunquist et al., 1973*), exceptional camouflage (*Suutari et al., 2010*) and preference for residing high up in the dense rainforest canopy (*Montgomery & Sunquist, 1978*), there is very little literature documenting the activity patterns of wild sloths. This is particularly the case for sloths in the *Choloepus* genus, for which there is only one set of data from wild animals (*Montgomery & Sunquist, 1978*; *Sunquist et al., 1973*). Nevertheless, sloth activity and its determinants are likely to be multifaceted due to their low-calorie folivorous diet (*Chiarello, 1998b*; *Nagy & Montgomery, 1980*) and extremely slow rate of digestion (*Foley, Engelhardt & Charles-Dominique, 1995*; *Gilmore, Da Costa & Duarte, 2001*; *Vendl et al., 2016*), which, together, critically limit rates of energy acquisition. In addition, sloths are considered to have an unusually low and variable body temperature compared to typical endothermic homeotherms (*Gilmore, Da Costa & Duarte, 2001*; *Goffart, 1971*), so

their metabolism (*Cliffe et al., 2018*), food intake (*Cliffe et al., 2015*) and consequently all aspects of energy use, are dependent upon the environmental conditions. It is thus expected that fluctuations in ambient temperature, and therefore sloth body temperature, will impose metabolic limitations on activity (*Cliffe et al., 2018*; *Giné et al., 2015*) as well as requiring animals to employ behavioural strategies to control rates of heat loss and gain. In line with this, wild sloths have been observed to exhibit thermoregulatory behaviours such as the extension/retraction of limbs while resting and varying their position within the canopy to manipulate levels of sunlight exposure (*Gilmore, Da Costa & Duarte, 2001*; *Urbani & Bosque, 2007*).

Data available on the activity of *Bradypus* sloths all report great variation, which has been explained by food availability, predator avoidance and climatic differences between study sites (*Castro-Vásquez et al., 2010*; *Chiarello, 2008*; *Gilmore, Da Costa & Duarte, 2001*; *Oliveira Bezerra et al., 2020*; *Pinder, 1985*; *Queiroz, 1995*; *Urbani & Bosque, 2007*). To date, only two studies have explored the relationship between ambient temperature and sloth activity (*Giné et al., 2015*; *Sunquist et al., 1973*). These two studies, along with (*Oliveira Bezerra et al., 2020*), also represent the only collections of nocturnal data, with other work either ignoring the dark phase entirely, or inferring levels of night-time activity by comparing the distances travelled diurnally to those travelled nocturnally (*Chiarello, 1998a*; *Chiarello, 2008*). This method likely misses a substantial amount of behaviour as displacement is considered to represent less than 50% of total sloth activity (*Castro-Sa, Dias-Silva & Barnett, 2021*), with individual animals showing small bouts of activity before repeatedly returning to the same resting place. Previous work is further limited by the same reoccurring problems, including small sample sizes, a lack of continuous behavioural and/or environmental data, and the obvious difficulties associated with making direct behavioural observations on a cryptic species which has a propensity for living high up in the canopies of dense, remote tropical rainforests (*Montgomery & Sunquist, 1978*).

Based on available data, *Choloepus* sloths are considered to be nocturnal (*Sunquist et al., 1973*), whereas *Bradypus* sloths show great variation. Some populations appear to show diurnal activity (*Oliveira Bezerra et al., 2020*; *Urbani & Bosque, 2007*), while others report nocturnal activity (*Castro-Vásquez et al., 2010*; *Pinder, 1985*; *Queiroz, 1995*), and others still report that they are cathemeral (*Castro-Sa, Dias-Silva & Barnett, 2021*; *Chiarello, 1998a*; *Giné et al., 2015*; *Sunquist et al., 1973*). It has been noted though, that the regions in which higher levels of diurnal activity are observed tend to be climatically cooler due to higher elevations, while warmer lowland areas are associated with an increased frequency of nocturnal movements (*Chiarello, 2008*; *Giné et al., 2015*). However, the influence of ambient temperature on the behavioural ecology of all sloth species is far from being understood, not least perhaps because other important activity modulators of tropical animals, *e.g.*, precipitation and wind speed (*Halle, 2000*), have not been placed in context with respect to the activity of wild sloths. A pattern that does emerge from the available literature, however, is that there is a high level of inter-individual variation in the amount of time spent active and in the 24-hour temporal distribution of activity, resulting in a distinct lack of synchrony between animals in a population (*Castro-Sa, Dias-Silva & Barnett, 2021*; *Chiarello, 1998a*; *Chiarello, 2008*; *Giné et al., 2015*; *Queiroz, 1995*). Individuals appear to

operate on their own unique patterns of activity (*Chiarello et al., 2004*) which has led to speculation that sloths may entirely lack a circadian rhythm (*Chiarello, 2008*; *Queiroz, 1995*; *Sunquist et al., 1973*). As sloths have a diverse array of both nocturnal and diurnal predators (including spectacled owls (*Pulsatrix perspicillata*) (*Voirin et al., 2009*)), big cats such as jaguars (*Panthera onca*) and ocelots (*Leopardus pardalis*), harpy eagles (*Harpia harpyja*) (De Miranda et al., 2020) and tayras (*Eira Barbara*) (Sáenz-Bolaños et al., 2018), cathemerality and asynchronous activity of individuals within the same population may also function as an effective predator avoidance strategy (*i.e.,* favouring unpredictability) (*Richardson et al., 2018*).

Understanding the temporal activity patterns of sloths and their ability to withstand environmental fluctuations is of growing importance when faced with a dramatically changing world. Climate change alongside with various anthropogenic pressures (land-use change, deforestation, agricultural intensification, and rainforest urbanisation) are threatening sloth populations throughout South and Central America, with both species of sloth in Costa Rica recognised as conservation concerns in the country (*Cliffe et al., 2020*; *Rodriguez-Herrera, Chinchilla & May-Collado, 2002*). In order to develop effective conservation plans, we must first understand how sloths are able to cope with changes in their environment, and how these changes impact the behavioural ecology of the species.

Here, we used micro data loggers attached to wild *Bradypus* and *Choloepus* sloths inhabiting a lowland rainforest on the Caribbean coast of Costa Rica to record, for the first time, their behaviour continuously over periods of days to weeks. We examined how environmental conditions impact their activity budgets and how all of these factors might relate to their uniquely low power lifestyle.

Considering the known link between sloth metabolism, digestion, and ambient temperature (*Cliffe et al., 2015*; *Cliffe et al., 2018*), we hypothesised that sloths would favour nocturnal movements and show increased activity on cooler days in our study location (a hot lowland rainforest). As sloths rely on behavioural crypsis as their primary method of predator evasion, we also hypothesised that sloth movement would be positively correlated with wind speed due to an increased ability to blend in with swaying rainforest branches in windy conditions.

## MATERIALS AND METHODS

### Ethics
This research was approved by the Swansea University Animal Welfare & Ethical Review Process Group (AWERP), and the Costa Rican government and associated departments (MINAE, SINAC, ACLAC) permit number; R-033-2015.

### Sample
Data was collected over a 17-month period from April 2014 until August 2015 from eight free-living adult *Bradypus variegatus* sloths (seven male, one female) and four adult *Choloepus hoffmanni* sloths (all female) (Table 1). The exact ages of these animals were not known due to the difficulty of approximating age in wild sloths (*Peery & Pauli, 2014*).

**Table 1  List of sloths sampled.** Includes species, sex, body weight, date of release following tagging with a Daily Diary logger (DD) and the total duration of the resultant dataset.

| Dataset # | Sloth | Species | Sex | Weight (Kg) | Date released | DD data (hh:mm:ss) |
|---|---|---|---|---|---|---|
| 1 | | | | 2.20 | 20/2/2015 | 91:52:16 |
| 2 | Bojangles (bv1) | *B.variegatus* | M | 2.50 | 04/3/2015 | 102:39:37 |
| 3 | | | | 3.20 | 09/7/2015 | 156:55:23 |
| 4 | | | | 3.20 | 25/7/2015 | 180:44:19 |
| 5 | Burrito (bv2) | *B.variegatus* | M | 4.80 | 24/4/2014 | 179:18:16 |
| 6 | | | | 4.70 | 07/8/2015 | 92:50:13 |
| 7 | Guillermo (bv3) | *B.variegatus* | M | 3.45 | 04/8/2015 | 55:10:21 |
| 8 | Quatro (bv4) | *B.variegatus* | M | 4.30 | 27/3/2015 | 184:38:12 |
| 9 | Spock (bv5) | *B.variegatus* | M | 3.90 | 15/4/2015 | 156:46:17 |
| 10 | Star (bv6) | *B.variegatus* | F | 3.80 | 20/11/2014 | 116:53:50 |
| 11 | | | | 4.00 | 11/2/2015 | 124:28:59 |
| 12 | | | | 3.90 | 17/2/2015 | 181:29:28 |
| 13 | | | | 3.90 | 09/5/2015 | 187:26:52 |
| 14 | | | | 4.00 | 18/5/2015 | 30:21:08 |
| 15 | | | | 3.90 | 06/6/2015 | 201:04:03 |
| 16 | Steve (bv7) | B.variegatus | M | 4.25 | 18/7/2015 | 32:50:39 |
| 17 | Valentino (bv8) | B.variegatus | M | 5.00 | 08/9/2014 | 169:51:41 |
| 18 | | | | 5.00 | 26/7/2015 | 54:53:55 |
| 19 | | | | 5.00 | 03/8/2015 | 55:46:44 |
| | **Total** | | | | | 2356:02:13 |
| 20 | Beckett (ch1) | *C.hoffmanni* | F | 8.20 | 16/11/2014 | 29:56:35 |
| 21 | | | F | 8.25 | 15/4/2015 | 46:37:35 |
| 22 | Lizz (ch2) | *C.hoffmanni* | F | 5.40 | 15/4/2015 | 11:59:53 |
| 23 | Walda (ch3) | *C.hoffmanni* | F | 7.30 | 10/3/2015 | 43:10:31 |
| 24 | Willa (ch4) | *C.hoffmanni* | F | 6.90 | 10/3/2015 | 74:20:01 |
| | **Total** | | | | | 206:04:35 |

However, all body weight measurements fell within the expected adult range for these species (*Gilmore, Da Costa & Duarte, 2001*).

## Study area

All sloths were from the same region of secondary forest surrounding the Sloth Sanctuary of Costa Rica (N 09°47′56.47″, W 082°54′47.20″). This forest is protected from development and has a high level of canopy connectivity, however, there are some anthropogenic stressors present in the area including human settlements, domestic dogs, and a major highway. There is a variety of other species present in the area including other arboreal folivores such as howler monkeys (*Alouatta caraya*), and predators including tayras (*Eira barbara*), spectacled owls (*Pulsatrix perspicillata*), boas (*Boa constrictor*), and ocelots (*Leopardus pardalis*). Notably, the sloth's primary diurnal predator, the harpy eagle (*Harpia harpyja*) is extinct in the region.
The study region maintains high and stable temperatures year-round with high levels of humidity and rainfall. Although there is minimal seasonal variation in temperature, four *Bradypus* and one *Choloepus* sloth were repeatedly tagged at different times of the year to investigate whether any seasonal variations in activity patterns could be identified (Table 1).

## Capture and tagging procedure

All *Bradypus* sloths were caught opportunistically by hand and equipped with tags (see below) without anaesthesia. *Choloepus* sloths were anaesthetised during capture using 1 mg/kg of ketamine (Ketamina 50®; Holliday-Scott, Buenos Aires, Argentina) and 0.008 mg/kg of dexmedetomidine (Dexdomitor®; Zoetis, Parsippany-Troy Hills, NJ, USA) administered intramuscularly. Anaesthesia was reversed prior to release using 0.008 mg/kg of anti-sedante (atipamezol; Antisedan®, Zoetis). Body mass measurements were taken for each individual at capture (E-PRANCE® Portable Hanging Scale (±0.01 g)).

Sloths were equipped with pre-calibrated Daily Diary (DD) data loggers (*Wilson, Shepard & Liebsch, 2008*), programmed to record 9 parameters (barometric pressure (mbar), external temperature (°C), relative humidity (%), tri-axial magnetometry (gauss), and tri-axial acceleration (g)) at a rate of 40 Hz. The data loggers were held within non-lubricated condoms containing 0.5 g silica gel desiccant to protect the electronics from moisture damage. DDs were combined with Very High Frequency (VHF) radio transmitters (Biotrack PIP3 VHF tag) within 3D-printed housings and attached *via* elastic, backpack-style harnesses, positioning the devices firmly on the upper back (Fig. S1) (*Cliffe et al., 2014*). The total weight of the backpack was 90 g.

## Post-release monitoring

After being equipped with a backpack, all sloths were released in the same location they were originally found. Backpacks were retrieved manually when the sloth was in an appropriate position for recapture close to the ground following daily checks. Due to the difficulty of recapturing *Choloepus* sloths, a small link of dissolvable film (Aquatics ROMEO) was braided into the harness of these animals which allowed the backpack to fall off when exposed to heavy rain.

## Weather data

Corresponding weather data was recorded for 997 h of *Bradypus* sloth data and 176 h of *Choloepus* sloth data to coincide with tag deployment periods. Ambient temperature (°C), humidity (%), wind speed (m/s), maximum gust speed (m/s) and rainfall (mm/15 min) were recorded every 15 min by a weather station (Davis Vantage Vue) mounted 17 m above the ground on the edge of the study site. Light intensity was measured using a HOBO Pendant® UA-002-64-Temperature/Light Data Logger (64K) mounted adjacent to the weather station.

## Data analysis

DD data were analysed using the computer software program DDMT (http://www.wildbytetechnologies.com/software.html). Standard metrics for looking at animal

behaviour (*Grundy et al., 2009*; *Shepard et al., 2008*) involving static and dynamic acceleration derivatives are difficult to apply to sloth data because these animals have very little dynamism in their movement (see later) although careful examination of the data from all channels indicated clear patterns from various behaviour categories. To ascertain this, and to provide a behaviour identification key, Daily Diary tags were attached to captive sloths and extensive observation undertaken which allowed us to equate sensor signals with 6 primary behaviours. These were: locomotion (climbing), climbing upwards, climbing downwards, grooming, resting, and sleeping (Figs. S2 & S3). Sensor signals of value included slow changes in the static acceleration and magnetometry signals combined with the pressure rate of change. Feeding could not be distinguished from resting due to leaf mastication generating no distinguishable body movements. Behaviours were broadly categorized into 'active' (including grooming and all climbing—Fig. S2) and 'inactive' (including sleeping and resting—Fig. S3) based on the Vector of the Dynamic Body Acceleration (VeDBA), a proxy for energy expenditure (*Gleiss, Wilson & Shepard, 2011*; *Wilson et al., 2006*; *Wilson et al., 2020*), with inactive behaviours typically producing a signal less than 0.03 *g*.

## Statistical analysis

All data were classified using 'expert' observers. The observers were trained using the calibration datasets and then given unknown samples from a calibration dataset to ascertain those behaviours were being correctly classified. Observers worked within DDMT (in-house software), specifically designed to enhance manual identification of Daily Diary datasets. Data from all sensors within a single tag and derivatives (VeDBA and rate of change metrics) were displayed as time-based parallel traces that were inspected over both time- and sensor resolution scales.

To determine overall activity budgets, the total percent time allocated to different behaviours was calculated for each dataset.

## Activity budgets

To ascertain whether *Bradypus* and *Choloepus* sloths differed in their overall levels of activity, the mean hourly percent time spent active was calculated and compared using a Wilcoxon rank-sum test. The significance of between-individual variation in the mean hourly percent time spent active was calculated using Kruskal-Wallis tests.

The significance of the difference in the total amount of time spent climbing upwards *vs* climbing downwards for all sloths combined was determined using a Welch's two-sample *t*-test. The total time spent climbing downwards (for all sloths combined) was compared across different times of the day using an analysis of variance (ANOVA) followed by a Tukey Honestly Significant Differences (Tukey's HSD) test. This was repeated for the total time spent climbing upwards. For these tests, the 24-hour day was broken down into 4 bins: morning (06:00–12:00), afternoon (12:00–18:00), evening (18:00–00:00) and night (00:00–06:00). The significance of the relationship between climbing upwards and light intensity was tested by using a point bi-serial correlation case of the Pearson's product-moment coefficient (as ''climbing up'' is a Boolean variable).

### Activity patterns and effect of the environmental conditions

To determine the effect of the environmental conditions, the mean hourly percent times spent active, mean hourly ambient temperatures, and mean hourly maximum wind speeds were calculated. Using this data, and assuming diurnal hours to be between 05:00 and 17:00, and nocturnal hours to be between 17:00 and 05:00, the daily percent times spent active diurnally and nocturnally were calculated for each sloth for each day that had corresponding weather data (four individuals totalling 29 days for *Bradypus* sloths and four individuals totalling 11 days for *Choloepus* sloths). Corresponding daytime and night-time ambient temperatures were also calculated.

The significance of the relationships between the percent time spent diurnally active and the mean diurnal temperature for both *Bradypus* and *Choloepus* sloths were determined using Spearman's Rank correlations. The significance of the relationships between the mean percent time spent nocturnally active and the mean daily and nightly ambient temperatures were determined using general linear models (GLM) using normal data that was centred and box-cox transformed (power transformation of 0.3608). The significance of the relationships between maximum wind speed and the mean percent time spent active for the day and night were tested using Spearman's Rank correlations.

In order to determine the significance of the difference in percent time spent active between different periods of the day for each sloth, the 24-hour day was again broken down into 4 bins: dawn (04:00–07:00), day (07:00–16:00), dusk (16:00–19:00) and night (19:00–04:00). The percent times spent active during each bin were compared using a Kruskal-Wallis test with a significance threshold of $\alpha = 0.05$, followed by a Dunn's *post hoc* test to identify which periods of the day differed from each other. Based on the significance level, sloths are identified as either nocturnal (significantly higher activity during the night compared to the day), diurnal (significantly higher activity during the day compared to the night) or cathemeral (no significant difference in activity throughout the day according to a Kruskal Wallis test).

## RESULTS

### Activity budgets

A total of 2,356 h of behavioural data was collected for *Bradypus* sloths and 206 h for *Choloepus* sloths. *Bradypus* sloths were inactive for a total of 85.5% of time, comprising 62.7% and 22.8% of time spent sleeping and resting, respectively. There was significant between-individual variation in their mean hourly percent time spent active ($\chi 2 = 57.1$, $df = 7, p\text{-value} < 0.001$), with total time spent in inactive behaviours ranging from 49.3–97.1% (Fig. 1 & Tables S1–S2).

*Choloepus* sloths spent a total of 72.6% of their time inactive, comprising 56.4% and 16.2% of their time sleeping and resting. There was no significant difference in the mean hourly percent time spent active between *Choloepus* and *Bradypus* ($W = 8742.5$, $N = 288$, $P = 0.433$), however *Choloepus* sloths also demonstrated significant between-individual variation ($\chi 2 = 25.0$, $df = 3, p\text{-value} < .001$), with total time spent in inactive behaviours ranging from 37.8–95.2% (Fig. 1 & Tables S1–S2).

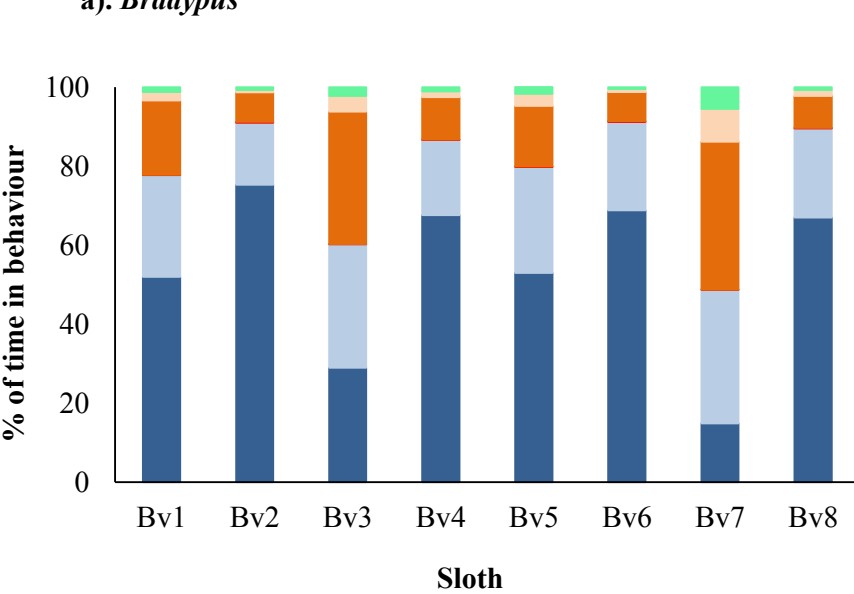

**a). *Bradypus***

**b). *Choloepus***

Legend:
- climbing down
- climbing up
- climbing
- grooming
- resting
- sleeping

**Figure 1** **Variation in the percent of time spent in different behaviors for eight wild *Bradypus* (A) and four wild *Choloepus* (B) sloths.** Each behaviour is denoted by a different colour.

Both *Bradypus* and *Choloepus* sloths spent a significantly higher proportion of total percent time climbing upwards (1.5% and 3.2% of time, respectively) compared to climbing downwards, where the equivalent values were (0.1% and 1.2% of time, respectively) ($p$-value = 0.044). Sloths spent more time making vertical movements within the canopy during nocturnal hours. This included significantly more time spent climbing downwards during the night compared either to the morning or afternoon ($p$-values = 0.001 and

0.039, respectively). Similarly, sloths spent a significantly higher proportion of total active time climbing upwards at night compared to the morning, afternoon, or evening (*p*-values = 0.02, 0.004, and 0.012, respectively). There was no significant correlation between the occurrence of upwards climbs and light intensity for all sloths ($r = -0.015$, *p*-value = 0.508).

## Activity patterns and the effect of environmental conditions

The mean and standard deviation in ambient temperature during the study period was of 26.3 ± 3.0 °C, ranging from an overall recorded minimum of 20.5 °C to an overall recorded maximum of 32.8 °C. There was no significant difference in the amount of time spent diurnally active and the mean ambient temperature during daytime hours (05:00–17:00) for *Bradypus* or *Choloepus* sloths ($r = 0.011$, $df = 27$, *p*-value = 0.956 and $r = 0.329$, $df = 9$, *p*-value = 0.323, respectively). However, *Bradypus* did spend significantly more time active nocturnally (17:00–05:00) on days which had cooler ambient daytime temperatures (05:00–17:00) ($\chi 2 = 2.672$, $df = 24,3$, *p*-value = 0.018), with individuals differing significantly in their response to this. Overall, there was also a strong significant negative relationship between the percent time *Bradypus* sloths spent nocturnally active and the ambient night-time temperature (17:00–05:00) ($\chi 2 = 4.233$, $df = 24,3$, *p*-value <0.001), with increased activity on colder nights (R squared = 0.731).

Maximum wind speed did not significantly affect the amount of time spent active for either *Choloepus* or *Bradypus* during either the day or the night.

Actograms reveal that all sloths were typically active in frequent small bursts lasting on average 5.52 ± 8.55 min, interspersed by longer resting and/or sleeping periods (Table S3 & Figs. S4–S15). When data for all 8 individuals was considered together, *Bradypus* sloths were significantly more active during both the daytime ($\chi 2 = 113.57$, $df = 3$, *p*-value < 0.001) and at night ($\chi 2 = -5.69$, $df = 3$, *p*-value < 0.001) compared to at dawn, and they were significantly more active during the day than they were at dusk ($\chi 2 = 3.86$, $df = 3$, *p*-value < 0.001). However, there was significant variation in the periodicity of activity between individuals (Table S4); some sloths significantly favoured nocturnal activity, some favoured diurnal activity, and some favoured a cathemeral pattern, with peaks of activity occurring during both the day and night (Fig. 2 & Figs. S4–S11). *Choloepus* sloths were cathemeral in their activity patterns overall, with no significant difference in the amount of time spent active between dawn, day, dusk, and night (Table S4). Only one individual (ch3) displayed a significant reduction in activity at dawn, with activity remaining equal throughout the rest of the diel (Fig. 2 & Figs. S12–S15).

## DISCUSSION

The activity budgets of both *Bradypus* and *Choloepus* sloths observed in this study are broadly similar to those previously reported for wild sloths in terms of percent time allocated for different behaviours and overall activity, and levels of variation between individuals in activity (*Castro-Vásquez et al., 2010*; *Chiarello, 1998a*; *Giné et al., 2015*; *Oliveira Bezerra et al., 2020*; *Sunquist et al., 1973*; *Urbani & Bosque, 2007*). The overall levels of sloth activity are also comparable to those observed in other, similar sized arboreal

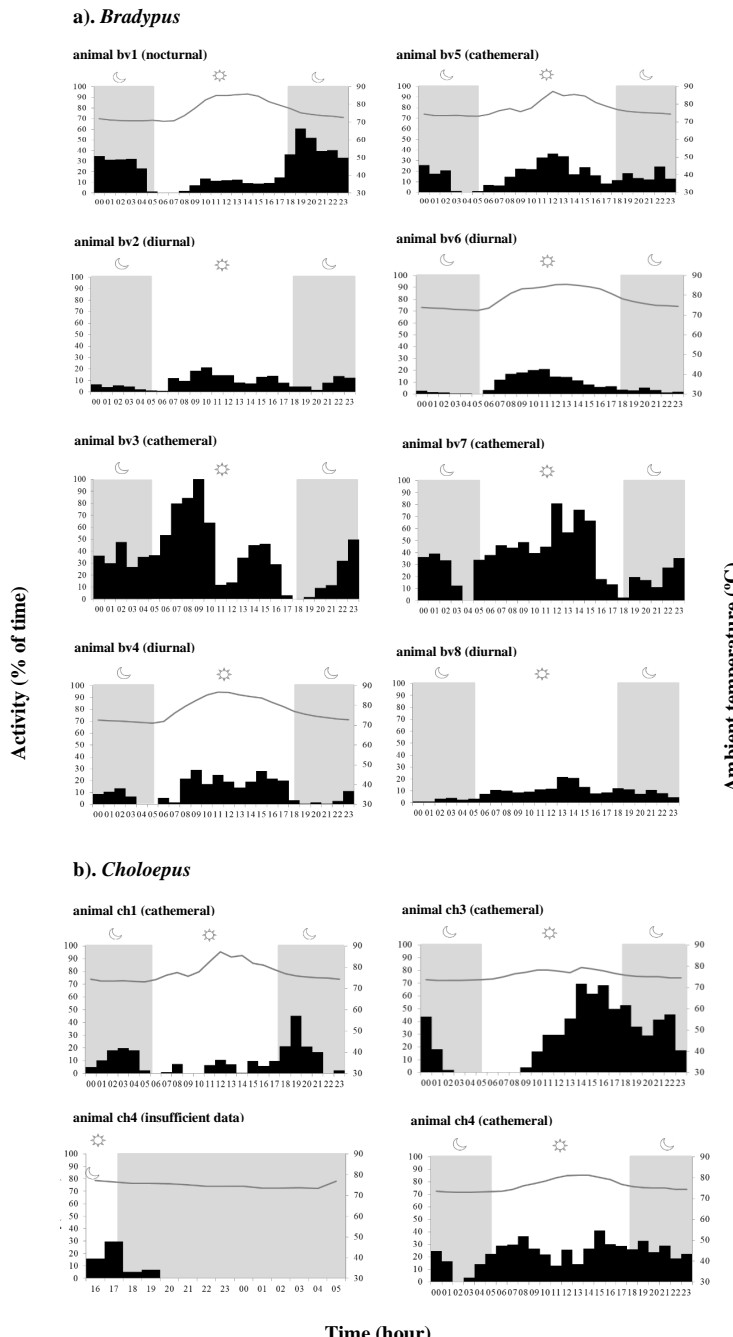

**Figure 2** Variation in the periodicity of activity and ambient temperature over 24 h for (A) *Bradypus* sloths and (B) four *Choloepus* sloths in Costa Rica to illustrate inter-individual variation. Dark phases are shown by grey shading. Because of the number of 0's in the data, the interquartile range varies between 0–93.3 and is not shown in the figure.

folivores, including the brown howler monkey (*Alouatta guariba*) (*Chiarello, 1993*) and koala (*Phascolarctos cinereus*) (*Ryan et al., 2013*). While observers could not accurately distinguish feeding from resting, feeding is thought to account for only a small portion of the sloth's daily activity budget (*Chiarello, 1998a*; *Giné et al., 2015*; *Urbani & Bosque, 2007*) and incurs little movement or energetic cost (as also indicated by our VeDBA data). The overall low levels of activity displayed by both *Bradypus* and *Choloepus* sloths are consistent with those expected for a folivorous mammal with a low metabolic rate (*McNab, 1978*; *McNab, 1982*; *Nagy & Montgomery, 1980*; *Vendl et al., 2016*).

All sloths were found to spend more time climbing upwards than downwards, suggesting that it may take a longer "period" to cover the same distance when climbing upwards due to the added effort of moving against gravity. This is particularly pertinent in sloths, with their low power usage, because the biomechanical power invested for climbing upwards is given by the rate of change of potential energy (where PE = mgh; m = mass, g = gravitational constant and h is the height difference) so that power usage is directly proportional to speed. Sloths were also found to spend more time making vertical movements within the canopy during the night than they did during the day, although overall levels of activity did not differ within nychthemeral cycle. This may represent an increased tendency to descend to the ground during the night, although this would seem surprising considering the increased threat of predation from nocturnally active large cats (*Harmsen et al., 2011*; *Huck, Juárez & Fernández-Duque, 2017*). Alternatively, it may represent a slower rate of movement during the night. The reason for this is unknown, however it is unlikely to be a consequence of reduced visual capabilities considering sloth eyes are only functional in extremely low light conditions (*Emerling & Springer, 2014*).

While *Choloepus* sloths were previously considered to be strictly nocturnal (*Sunquist et al., 1973*), the sloths in this study were cathemeral in their activity patterns, with only one individual showing a marked decrease in activity at dawn. Our ability to be definitive about the activity patterns of *Choloepus* sloths is limited by our small sample size, but given the backdrop of almost no literature at all on the behavioural ecology of wild two-fingered sloths, our data represent a substantial contribution to current knowledge (*Alvarez, Sanchez & Carmona, 2004*; *Chiarello, 2008*; *Sunquist et al., 1973*).

*Bradypus* sloths were cathemeral in their activity patterns when all individuals were considered together, with a curious marked depression in activity at both dawn and dusk (*Sunquist et al., 1973*). Within this broad classification, however, there was significant variation in the temporal distribution of activity between individuals, with some clearly favouring nocturnality, and some favouring diurnality (Fig. 2). High levels of inter-individual variation in patterns of activity for sloths inhabiting the same portion of forest, at the same time, demonstrate a distinct lack of synchronicity within the same population. While the different ages or physiological states of the "studied" animals (which we could not determine) may have influenced activity, this pattern is in accord with the existing literature and suggests that the reported variation in the periodicity of activity between different regions may not be entirely due to climatic differences, and rather could simply reflect the high levels of variation that exist between all sloths anyway (*Castro-Sa, Dias-Silva & Barnett, 2021*; *Castro-Vásquez et al., 2010*; *Chiarello, 1998a*; *Chiarello, 2008*; *Chiarello et*

*al., 2004*; *Gilmore, Da Costa & Duarte, 2001*; *Giné et al., 2015*; *Sunquist et al., 1973*; *Urbani & Bosque, 2007*). Actograms reveal a high level of intra-individual variation in the total percent time spent active by the same *Bradypus* sloths on different days, demonstrating what appears to be a lack of any regular activity pattern within individuals as well as between individuals. Some of the variation in individual levels of nocturnal activity can be related to differences in ambient temperature, with sloths showing increased levels of activity during both colder nights and nights that followed colder days. Additionally, the sporadic patterns of day-to-day activity may also reflect the individual animal's requirement to thermoregulate or travel to find food or a mate. Sloths have highly specific feeding preferences for young leaves, encompassing approximately 7–19 tree species based on the secondary metabolites present in the leaves (*Chiarello, 1998b*; *Chiarello, 2008*). This preference is passed down from mother to infant and subsequently differs substantially between individuals (*Chiarello et al., 2004*; *Montgomery & Sunquist, 1978*). *Bradypus* sloths move between preferred feeding or 'modal' trees in a cyclic rotation, often spending consecutive days in the same tree before moving to the next. It may be that sloths are more active on the days which involve changing tree; however more work would be required in order to test this hypothesis.

It is possible that sex and reproductive-related differences may contribute to the observed high levels of inter-individual variation in activity, however the uneven sex-distribution of our sample limits our ability to account for this. The oestrus cycle for female three-fingered sloths is still unknown, and there is much speculation as to the presence of a mating season to ensure birth coincides with favourable climatic conditions; however, there are many conflicting arguments over this (*Gilmore, Da-Costa & Duarte, 2000*; *Lara-Ruiz & Chiarello, 2005*; *Martins Bezerra et al., 2008*). It seems likely that reproductive seasonality may differ by region based on phenological and climatic differences (Dias et al., 2008). From what we know about sloths in the study region (R Cliffe, pers. obs., 2009–2022) there is no reproductive season—probably due to the absence of any significant wet and/or dry seasons—and both *Bradypus* and *Choloepus* sloths are thought to reproduce year-round.

While there is still much to learn about the drivers of sloth activity, it appears that the irregular observed patterns are a consequence of complex interactions between their natural endogenous rhythms (such as exist), their physiology, and the environmental conditions. Inter-individual differences in activity have historically been ignored or considered to be of little scientific value, however such variances likely form an adaptive flexibility of substantial biological importance (*Slater, 1981*).

Although uncommon, cathemeral activity patterns with high levels of individual variation have been previously observed in other mammals and in some cases are thought to be an effective method of evading or confusing predators (*Pepin & Cargnelutti, 1994*). Avoiding detection by predators is likely to be a factor of high importance for sloths as, due to their low metabolic rate, they lack the ability to run away or defend themselves if detected (*Voirin et al., 2009*). Another common benefit of cathemeral activity patterns in mammals is the avoidance of competition (*Curtis & Rasmussen, 2006*). While male *Bradypus* sloths are known to engage in territorial disputes over access to females (*Greene, 1989*; *Pauli, Peery & Festa-Bianchet, 2012*), it seems that there is no competition over access to resources since

there are few vertebrates that subsist primarily on leaves. Multiple sloths are often observed residing in the same tree, and large overlaps in home ranges are common (*Pauli, Peery & Festa-Bianchet, 2012*). This is likely feasible due to an abundance of food, high levels of individual specificity in diet choice (*Chiarello, 1998b*; *Chiarello et al., 2004*), and low levels of food consumption due to a slow digestive rate (*Cliffe et al., 2015*; *Foley, Engelhardt & Charles-Dominique, 1995*; *Montgomery & Sunquist, 1975*; *Nagy & Montgomery, 1980*). In line with this, any potential benefits relating to metabolic demands, or the temporal distribution of food are unlikely. There is no seasonal variability in the availability of food, and the feeding trees favoured by sloths tend to be asynchronous in the production of new leaves, thereby ensuring a constant supply of new leaves year-round (*Chiarello, 2008*).

It would therefore seem that the primary benefit of a cathemeral activity pattern for sloths is the flexibility to exploit favourable environmental conditions, without any added risk of predation. This would also provide a reasonable explanation for the substantial day-to-day variation in activity within individuals. Sloths, having at least partially sacrificed their capacity for adaptive thermogenesis, have a body temperature and metabolic rate which fluctuate widely (by mammalian standards) based on the ambient conditions (*Cliffe et al., 2018*; *Irving, Scholander & Grinnell, 1942*; *Nagy & Montgomery, 1980*). This, combined with a low caloric diet, means that sloths have very little energy at their disposal. In order to maximise their energy budget and considering their need for movement is based on factors which are rarely time-limited, it would therefore seem beneficial to delay activity until the environmental conditions are at their most favourable.

Flexible activity patterns in response to abiotic factors have also been observed in other members of the Xenarthra magnorder, including the giant anteater (*Myrmecophaga tridactyla*) (*Di Blanco, Spørring & Di Bitetti, 2017*) and the nine-banded armadillo (*Dasypus novemcinctus*) (*Norris, Michalski & Peres, 2010*), as well as in other sedentary arboreal mammals such as the koala (*Phascolarctos cinereus*) (*Ryan et al., 2013*). However, it does seem surprising that the *Bradypus* sloths in this study were significantly more active on colder nights, especially considering the depressing effect that low temperatures have on sloth metabolic rate (*Cliffe et al., 2018*). It has been previously theorised that sloths may increase nocturnal activity in warmer regions (*Chiarello, 1993*; *Giné et al., 2015*; *Oliveira Bezerra et al., 2020*), and so the observed pattern may be related to the comparatively warmer climate of this study region (*Giné et al., 2015*).

Why then, is there a tendency for *Bradypus* sloths to reduce activity at dawn and dusk? Predatory animals including medium and large cats often display crepuscular patterns of activity in order to exploit the vulnerability of early-starting nocturnal (or diurnal) prey species that are operating at their visual limits in terms of light intensity (*Daly et al., 1992*; *Foster et al., 2013*; *Harmsen et al., 2011*; *Helfman, 1986*). It would therefore seem beneficial for a species such as the sloth to minimise movement during the twilight hours when a large proportion of its predators may be at their most active (the predator-avoidance hypothesis) (*Carrillo, Fuller & Saenz, 2009*; *Shamoon et al., 2018*; *Sih & McCarthy, 2002*).

Due to the extreme range of light intensities experienced by cathemeral animals, the sensory systems of these species are exposed to different selection pressures compared to those that restrict activity to nocturnal or diurnal periods. For example, while diurnal

mammals tend to have morphological adaptations of the eye to maximise visual acuity, and nocturnal mammals to maximise visual sensitivity, cathemeral mammals typically maintain a compromise between the two extremes (*Peichl et al., 2019*; *Veilleux & Lewis, 2011*). Sloths are rod monochromats (completely lacking cone cells in the retina) (*Emerling & Springer, 2014*) and so appear more specialised to exploit the nocturnal conditions than the typical cathemeral mammal. This condition results in total colour blindness and represents an extreme adaptation to dim light conditions. Sloths consequently have very poor visual acuity when light levels are low and are completely blind when exposed to bright light, so that they probably only use vision at dawn, dusk, and during the night. Regardless, with their slow and deliberate style of locomotion, sloths have little need for high levels of visual acuity. In fact, they have little need for any level of high-speed processing, and as such, have a basal pattern of neural organisation and slow neuromuscular responses (*Espírito Santo Saraiva & Magalhães Castro, 1975*; *Gilmore, Da-Costa & Duarte, 2000*; *Toole & Bullock, 1973*). Considering that the metabolic cost of neural processing is high (*Laughlin, De Ruyter van Steveninck & Anderson, 1998*), and the sloth lifestyle has little requirement for well-developed sensory or reactive abilities, a reduction in neural processing would therefore substantially aid in reducing energetic demands. With the sloths' dependence on sensory and neural function minimised, the selective pressures exerted by circadian differences in light intensity become insignificant and cathemeral activity patterns are thus a feasible option.

## CONCLUSIONS

This study represents the first continual collection of behavioural data for both *Bradypus* and *Choloepus* sloths and shows them both to be cathemeral in their activity, with high levels of between-individual and within-individual variation in the amounts of time spent active, and in the temporal distribution of activity over the 24-hour cycle. *Choloepus* sloths were previously considered to be exclusively nocturnal and so the results of this study represent a substantial contribution to current knowledge on the activity of this species. *Bradypus* sloths were found to show increased nocturnal activity on colder nights and on nights following colder days, which may be due to the comparatively warmer climate of this study region (*Giné et al., 2015*). Our results further demonstrate a distinct lack of synchronicity between sloths within the same population, and we suggest that the previously reported variation in sloth activity patterns between different regions may not be exclusively due to climatic differences and could also reflect the high levels of variation that exist between all sloths anyway. Finally, we hypothesise that cathemerality with high levels of between-individual and within-individual variation in activity provides sloths with the flexibility to exploit favourable environmental conditions whilst also reducing the threat of predation.

## ACKNOWLEDGEMENTS

We thank the Sloth Sanctuary of Costa Rica for allowing us to conduct this research on their property and their advice, and Dr. Francisco Arroyo for his veterinary and logistical assistance throughout data collection.

### Funding

This research was funded by donations to an Indiegogo crowdfunding campaign and the Sloth Conservation Foundation. The funders had no role in study design, data collection and analysis, decision to publish, or preparation of the manuscript.

### Grant Disclosures

The following grant information was disclosed by the authors:
Indiegogo crowdfunding campaign and the Sloth Conservation Foundation.

### Competing Interests

The authors declare there are no competing interests. The authors are not aware of any competing interests that the Indiegogo crowdfunders and Sloth Conservation Foundation donors may have.

### Author Contributions

- Rebecca N. Cliffe conceived and designed the experiments, performed the experiments, analyzed the data, prepared figures and/or tables, authored or reviewed drafts of the article, and approved the final draft.
- Ryan J. Haupt analyzed the data, authored or reviewed drafts of the article, and approved the final draft.
- Sarah Kennedy performed the experiments, authored or reviewed drafts of the article, and approved the final draft.
- Cerys Felton performed the experiments, authored or reviewed drafts of the article, and approved the final draft.
- Hannah J. Williams analyzed the data, authored or reviewed drafts of the article, and approved the final draft.
- Judy Avey-Arroyo conceived and designed the experiments, authored or reviewed drafts of the article, and approved the final draft.
- Rory Wilson conceived and designed the experiments, analyzed the data, authored or reviewed drafts of the article, and approved the final draft.

### Animal Ethics

The following information was supplied relating to ethical approvals (*i.e.*, approving body and any reference numbers):

This research was approved by the Swansea University Animal Welfare & Ethical Review Process Group (AWERP), and the Costa Rican government and associated departments (MINAE, SINAC, ACLAC) permit number; R-033-2015.

### Field Study Permissions

The following information was supplied relating to field study approvals (*i.e.*, approving body and any reference numbers):

The Sloth Sanctuary of Costa Rica.

### Data Availability

The raw data exports from the software program DDMT is available in the Supplemental Files.

### Supplemental Information

Supplemental information for this article can be found online at http://dx.doi.org/10.7717/peerj.15430#supplemental-information.

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
