# Peer review of "The behaviour and activity budgets of two sympatric sloths; Bradypus variegatus and Choloepus hoffmanni"

_PeerJ, doi:10.7717/peerj.15430_

## Round 0.1 · original submission · Major Revisions

The referees have reviewed your manuscript carefully and recommended some modifications on the manuscript before its further processing. Hence, the decision “Minor revision” was taken for your submitted manuscript.

The referee would like to see easily the modifications made to your manuscript in the revised version. Therefore, I invite you to respond to the referee(s)' comments and revise your manuscript carefully.

Do not forget to highlight ALL the changes you make using track changes.

Please provide also an answer/report to the referee(s)’ comments, which summarizes the changes you have made IN the manuscript itself. The answer/report to the referee(s) may also include any other response that you want the editor and the reviewer(s) to note. You should submit the answer/report to the referee(s)’ comments as a separate document.

Thank you for submitting your manuscript to PeerJ and giving us the opportunity to consider your work.

We look forward to receiving your revision.

·

Basic reporting

The paper is clearly written with sufficient background and appropriate references. However Fig3 is a replication of some results from Fig 2. It is suggested to replace Fig 2 and 3 by one figure showing results for each individual apart with annotation of its classification (diurnal, nocturnal or cathermeral), to better outline the individual responses and eventually cathemerality
In Table S4, please mention what (N) does refer to?

Experimental design

The experimental design included (7 males and 1 female) Bradypus variegatus and (4 females) Choloepus hoffmanni to compare their activities, but did not present sex- related behavioral differences, particularly during the reproductive season. It is important to highlight such sexual dimorphism intra- and inter- species. There is great variation in activity (climbing) during males’ fight for mating (see, Greene, H. W. (1989). Agonistic behavior by three-toed sloths, Bradypus variegatus. Biotropica, 369-372.) that should be considered particularly for Bradypus specie (since the studied population includes both female and males).
Because of such sex – behavioral dimorphism, it is inappropriate to compare activities between the two populations (heterogeneous for sex distribution). Results of each specie should be separately described.
The Data analysis present different timing (15 minutes, Hour, day/night, and 4 inter-phases of the day) considered to evaluate the studied parameters. It is more appropriate to consider a unified time unit, e.g., activity per hour.
Throughout the study period (2014 – 2015) there should be incidence of some specific climatic changes (hurricanes, storm, etc), particularly from May to September. If any did occur, it should be mentioned and its effect on the experiment should be discussed.
The measurement of time spent in moving (climbing up or down) did not reflect the real energy spent by individuals. The frequency of movement (e.g. number of climbing up each 15 min), distance or speed of activities should be added to inform about the pragmatic activity’s budget.
.

Validity of the findings

The data was collected at different periods of the year (and eventually different years). To standardize conditions (for more rigorous statistic), it is suggested to use only data collected in days from the same period of the year and with closer climatic conditions
Results reported in L-314 to L-317 referring to variation within colder nights (“with increased activity on colder nights”) are not supported by any statistical analysis. Add the respective statistics

Additional comments

The paper describes the activity of two sloths’ species, based on time spent active. The studied matter is very interesting, especially in wild life and falls within the scope of the journal and merits its publication after a major revision. As Sloths have a daily sporadic activities, it is better to report results for each followed individual to cut with the diurnal- nocturnal - cathemeral events.

Reviewer 2 ·

Basic reporting

no comment

Experimental design

no comment

Validity of the findings

no comment

Additional comments

Ms. Ref. No.: The behaviour and activity budgets of two sympatric sloths; Bradypus variegatus and Choloepus hoffmanni (#77288)

Reviewers' comments:

Overview:

This study presents the interesting results of the behaviour and activity budgets of two sympatric sloths in South and Central America Bradypus variegatus and Choloepus hoffmanni two understudied species, In fact, only two studies have explored the relationship between ambient temperature and sloth activity (Sunquist et al. 1973) (Giné et al., 2015), probably due to the ecology and restricted distribution of these two species. The ms is a sound paper, which is well presented. Except that, I have some remarks that will have to be taken into account by the authors to improve their article.

Introduction section is excessively long; it would be interesting to reduce the number of paragraphs, because there is the repetition of certain details! Please keep only the relevant data for this study.

L 163 Sample and study site:
In my opinion, it will be important to separate this pg into two subtitles (Study area / sampling procedures..).
L164 Data was collected between April 2014 and August 2015!!! this sentence not clear (sampling was carried during a year ? or two months? Please clarify this information which is very useful for your study.

Figures caption:
i have noted that the paper contains a relativly big number of illustrations. I suggest to reduce the number of figures, keeping the most useful ones.

However, these comments are of minor importance and I recommend publication of the manuscript.

Annotated reviews are not available for download in order to protect the identity of reviewers who chose to remain anonymous.

---

## Round 0.2 · Minor Revisions

Dear Dr. Cliffe,

Thank you for your submission to PeerJ.

I am writing to inform you that your manuscript - The behaviour and activity budgets of two sympatric sloths; Bradypus variegatus and Choloepus hoffmanni - has been Accepted for publication.

Congratulations!

·

Basic reporting

the article present some language mistakes that should be checked.
L68 : correst patters « patterns »
L116. A pattern which does .. replace by A pattern that does ..
L121. Correct by : to speculation that sloths may entirely lack a circadian rhythm ..
L130: correct: change alongside with various ..
L285-286: correct : sloths spent a significantly higher proportion of total active time climbing upwards
L289-290: correct: significantly more time spent in climbing downwards during the night compared either to the morning or afternoon
L296: correct: The mean and standard deviation in ambient temperature during the study period was of 26.3 ± 3.0°C,
L325-326: correct : in terms of percent time allocated for different behaviours and overall activity, and levels of variation between individuals in activity
L338: correct: take longer “period” to cover
L339: add (is): This is particularly pertinent in sloths
L344: add within nycthemeral cycle: overall levels of activity did not differ within nycthemeral cycle.
L362: correct: physiological states of the “studied” animals (which we..
L389: correct: It likely seems that reproductive seasonality
L395: “patterns observed” replace by “observed patterns

Experimental design

The study is very intreesting and present relevant results concerning the behaviour of sloths.
the methodology sound meaningful and rigorous investigation was performed to answer the propsed subject.

Validity of the findings

The discussion is specultive and should be summarized. In effect, the authors brought large suggestive discussion concerning the relationship between predation and sloths' activity. This matter was not studied (no results were given concerning such issue) in this work. Thus it is plausible to summarize this part of discussion.

---

## Round 0.3 · accepted · Accept

Dear Dr. Cliffe,

Thank you for your submission to PeerJ.

I am writing to inform you that your manuscript - The behaviour and activity budgets of two sympatric sloths; Bradypus variegatus and Choloepus hoffmanni - has been Accepted for publication.

Congratulations!